# OpenReview forum: "Bias and Volatility: A Statistical Framework for Evaluating Large Language Model's Stereotypes and the Associated Generation Inconsistency"
_NeurIPS.cc/2024/Datasets_and_Benchmarks_Track — NeurIPS 2024 Track Datasets and Benchmarks Poster_

### Official Review · Reviewer_erKg · 2024-07-24
**Large benchmark of models, and proper metric definition**

**Rating:** 7
**Confidence:** 5
**Correctness:** Yes.
**Clarity:** Yes

**Review:**

# Summary
In this manuscript, the authors propose the Prejudice-Volatility Framework (PVF) a framework for studying bias and societal discrimination of LLMs. The manuscript defines discrimination as comprised of a base prejudice, e.g., against one group, and a volatility induced by different contextualizations of the prejudice.

# Strong points
1. Varied analysis of several models, and of the correlations of several variables.
2. Strong approach
# Weak points
1. No major weak points, some minor observations in section # Observations

# Observations
- When setting the unbiased model likelihoods, shouldn't $p^∗_{y \mid x} (c) = \dfrac{1}{\mid V \mid}$, where $V$ is the vocabulary? Why is the model constrained to only predict the given tokens?
- In "The Discrimination Risk Criterion J" isn't this limited to the chosen group $x$, as it is not sampled? If so, it could help to state this.sse
- $r^p_x$ may be better expressed as $r^C_x$, as it is the context $C$ varying.
- The elimination process ii) (Sec. 3.2) could be expanded upon: how are such sentences identified?
- Personally, I think scaling to 1000 the scores makes them less, rather than more, clear.
- "We fine-tuned LLama2 on toxic data" How was this toxic dataset generated?
- In figure 10a, why do most distributions have stop before risk 1?

# Soundness
4.

# Typos & related
- "Therefore, the token probabilistic nature causes LLM’s inconsistency in stereotype." Unclear sentence.

**Strengths:**

# Strong points
1. Varied analysis of several models, and of the correlations of several variables.
2. Strong approach

**Additional Feedback:**

# Observations
- When setting the unbiased model likelihoods, shouldn't $p^∗_{y \mid x} (c) = \dfrac{1}{\mid V \mid}$, where $V$ is the vocabulary? Why is the model constrained to only predict the given tokens?
- In "The Discrimination Risk Criterion J" isn't this limited to the chosen group $x$, as it is not sampled? If so, it could help to state this.sse
- $r^p_x$ may be better expressed as $r^C_x$, as it is the context $C$ varying.
- The elimination process ii) (Sec. 3.2) could be expanded upon: how are such sentences identified?
- Personally, I think scaling to 1000 the scores makes them less, rather than more, clear.
- "We fine-tuned LLama2 on toxic data" How was this toxic dataset generated?
- In figure 10a, why do most distributions have stop before risk 1?

# Soundness
4.

# Typos & related
- "Therefore, the token probabilistic nature causes LLM’s inconsistency in stereotype." Unclear sentence.

**Documentation:**

Yes

**Limitations:**

Yes.

**Opportunities For Improvement:**

The manuscript could be clarified a bit (see #Observations). No major weak points, some minor observations I've already highlighted.

**Relation To Prior Work:**

Yes

**Summary And Contributions:**

In this manuscript, the authors propose the Prejudice-Volatility Framework (PVF) a framework for studying bias and societal discrimination of LLMs. The manuscript defines discrimination as comprised of a base prejudice, e.g., against one group, and a volatility induced by different contextualizations of the prejudice.

---

> ### Author Rebuttal · Authors · 2024-08-16
>
> Thank you for your insightful observations and suggestions. It’s rewarding to learn that you view our discrimination assessment framework PVF, which identifies prejudice as the foundational bias and acknowledges the volatility introduced through the contextualization of prejudice, as a significant benchmark of models and a strong approach with proper metric definition. We have addressed your observations in order and hope our responses clarify your concerns.
>
> **Observation 1: Token Constraints of Unbiased Model Likelihood Definition**
>
> Thank you for your meticulous observation. We constrain the model’s predictions to a specific vocabulary related to the examined social discrimination to guarantee the correctness of PVF’s estimation of LLM’s discrimination risk. Indeed, if the PVF approach includes vocabulary that is unrelated to the attribute of the examined social discrimination, we will get an incorrect probability about the risk that applying the LLM causes a social-discrimination event.
>
> For example, when evaluating gender discrimination, suppose that in the use of gender-neutral terms,  $p$ predicts a 0.3 probability for male-related words, 0.3 for female-related words, and 0.4 for neutral words. In this scenario, model $p$ does not exhibit any stereotype. However, if $p*$ predicts a 0.2 probability for male-related words, 0.2 for female-related words, and 0.6 for neutral words, following Equation 1, the model would exhibit a 0.5 stereotype score for both males and females, which is unreasonable. Conversely, when we limit predictions to gender-related words, Equation 1 shows that the model exhibits a stereotype score of 0 for both males and females, more accurately reflecting the model's discriminatory tendencies.
>
> Furthermore, restricting predictions to a specific vocabulary helps avoid introducing noise from inaccurate predictions of unrelated words, which is analogous to how top-p or top-k sampling methods improve generated outputs.
>
> In selecting the vocabulary, we referred to sociological theories on discrimination and relevant linguistic research to ensure that our approach accurately reflects biases within the model. The conditional probabilities within a specific selection range can serve as an effective measure of stereotype strength. We will clarify this in the revision and provide a more detailed explanation of the theoretical basis for our choice of vocabulary.
>
> **Observation 2: Specification of Discrimination Risk Criterion $J$**
>
> Your understanding is correct: the computation of the discrimination risk criterion $J$ is specifically defined for the group $X$ determined during the initialization phase. Additionally, $J$ should be understood as a function of the conditional probability $p^*_{y|x}$, where $x$ serves as the evidence. We will consider clarifying this point in the revision.
>
> In the paper, we used gender discrimination in professions as an example. However, any other type of discrimination and language can be evaluated using the same criterion $J$. We have included the corresponding supplementary results for other type of discrimination and language in the additional PDF.
>
>
> **Observation 3: Prejudice Risk Notation**
>
> The determinants of model prediction include the context $c$ and the model $M$, as implied in Equation 1. We use $r_x^p$ to denote the prejudice risk against group $x$ across all sub-categories of attribute $Y$, where $p$ refers to "prejudice."
>
> **Observation 4: Context Sentence Identification and Elimination Process**
>
> Context templates $C$, augmented with variable $X$, are designed to facilitate coherent LLM inference and effectively assess the LLM's perspective regarding social group $X$. Our initial step for context mining involves filtering out sentences that lack simultaneous occurrences of words from both $X$ and $Y$ categories through keyword filtering. To avoid attribute-indicative adjectives or adverbs disturbing the semantics, we only retain core sentence components—mainly phrases or clauses that communicate the primary idea—and track their occurrences. We utilize established natural language processing toolkits for sentence parsing, eliminating all non-essential content, to extract these key sentence structures. Further clarification will be provided in the final version.
>
> Specifically, models identified as the most biased consistently exhibit the highest levels of bias across different data sources, while the least biased models maintain their ranking. This consistency suggests that the context selection method is indeed valid. The corresponding results have been included in the supplementary PDF.
>
> **Observation 5: Measurement Value Scaling**
>
> We will adopt the original values, as you suggest.
>
> **Observation 6: Toxicity Dataset Clarification**
>
> We employ the Toxic Tweets Dataset [Iyer, 2021], the Offensive Language Dataset [Davidson et al., 2017], and the Toxicity Dataset [Surge-ai, 2021] as our dataset for toxicity fine-tuning. In the revision, we will explicitly reference this in the main text.
>
> **Observation 7: Figure 10a Distribution**
>
> Figure 10a presents the original measurement values, not scaled by a factor of 1000 as in Table 1, potentially leading to confusion. To ensure clarity, we will uniformly apply the original values in the revision.
>
> **Unclear Sentence**
>
> Thank you for pointing it out. We will revise it to: "Therefore, the stochastic nature of LLM response sampling strategy causes inconsistency in their exhibited stereotype."

---

### Official Review · Reviewer_pUWU · 2024-07-24
**Prejudice and Volatility: A Statistical Framework for Measuring Social Discrimination in Large Language Models**

**Rating:** 6
**Confidence:** 4

**Review:**

The paper is well written and easy to follow. The problem statement is of utmost significance considering the wide use of LLMs in all spheres. The idea of dividing the stereotype distribution is novel to the best of my knowledge.

The illustrations in  Figures 2, 3 and 4 are difficult to comprehend. A suggestion is to make them simpler and clearer by removing the large chunks of information that has been tried to club into individual figures. The selection process in Figure 4 is not clear. The coherence between lines 149-154 and Figure 4 is difficult to follow.

Scale of R^v in Table 1 and that of (R, R^p) are anyway not similar from the definitions. The idea of scaling all three of them to 1000 is not quite clear. The range of R and R^p can be similar while that of R^v should be different and smaller.

Table 1: R^c -> R^v

For Section 4.2, besides the overall prejudice risk across all occupations, it will be interesting to have some empirical results across different professions like nurse, stylist, dietician that were exceptions for most of the models (as mentioned in lines 205-206).

It will be good to cite the used toxicity dataset in line 209 besides the Appendix. Also, excerpts from the dataset used for Table 1 and 2 may be provided for clarity. Clarifying the data used for the main results in Section 4.1 will be helpful.

The x-axis in Figure 9 should be clarified. If it's income group, how has it been categorised will be good to clarify in the main paper.

**Strengths:**

S0: The nuanced way to disentangle the notion of `discrimination' into prejudice and volatility is very well thought of. Regardless of the final decision of the review process, I must appreciate this distinction. It is very much analogous to how recently behavioral psychologists are trying to disentangle Biase from Noise.

S1: The paper deals with an important problem in the context of LLMs.

S2: The work tries to draw an analogy with human discrimination while proposing the different measurements for stereotype distribution.

S3: The experimental findings are intriguing.

**Additional Feedback:**

Explaining how are the proposed metrics standing out from the metrics defined in the existing works will be a valuable addition. For example, Gallegos, Isabel O., et al. "Bias and fairness in large language models: A survey." Computational Linguistics (2024) provides a taxonomy of metrics for measuring bias or discrimination. Are all of them measuring normal discriminatory behaviors in LLMs? A discussion on the existing metrics or some empirical results might be a valuable contribution for this work.

**Clarity:**

The paper is well written with some required clarifications which I have highlighted in different parts in the previous sections of the review.

**Correctness:**

Yes, the claims are correct. However, the dataset construction has to be elucidated more for clarity.

**Documentation:**

The dataset construction for the experiments need to be discussed in greater details.

**Ethics:**

NA.

**Limitations:**

There is no mention of limitations in the paper. The authors may reflect on the point whether these probability based metrics are sufficient in measuring the discrimination since they are highly dependent on the provided templates/context.

**Opportunities For Improvement:**

The experimental setup can be improved in terms of writing. Information on datasets used for particular experiments are not clear.
Figures can be made more clear,especially Figure 10. The takeaway from Section 4.5 is also not quite clear. It maybe fine to drop this section while explaining the other results and illustrations in a mroe succinct way.

**Relation To Prior Work:**

Yes.

**Summary And Contributions:**

The authors propose a Prejudice-Volatility framework (PVF) that define the discrimination risk of LLMs from two aspects - prejudice risk arising from system bias and volatility risk arising from inconsistencies in generated outputs. Their measures stand out from the existing works that only look at the discriminatory behaviors of the LLMs that might not be sufficient. The main contribution of this paper is the mathematical definition of stereotype distribution in LLMs. The explanation provided behind the intuition of prejudice and volatility in E.2 is appreciated.

---

> ### Author Rebuttal · Authors · 2024-08-16
>
> We greatly appreciate your constructive feedback. It's heartening to learn that you recognize our discrimination assessment framework PVF, which evaluates discrimination based on stereotype distribution and mathematically separates discrimination into prejudice and volatility, as a novel and distinctive contribution. Your positive remarks about our intuitive explanation of this separation, PVF's foundation in behavioral psychology, and our findings are highly valued. We organize our responses to your questions in the order they are presented and hope they address any concerns you may have.
>
> **Figure Interpretation**
>
> We appreciate the reviewer’s suggestion to simplify the figures for better clarity and understanding. In the revised version, we will provide a more precise explanation of the figures. Below, we briefly describe each figure.
>
> Figure 2 demonstrates how an LLM's bias regarding a particular profession fluctuates across different contexts and illustrates the underlying intuition behind the PVF framework.
>
> Figure 3 offers an intuitive graphical representation of the variables we define, which include (from left to right): $p^*$, $s_{y|x}$, $p_{y|x}(c)$, $J_x(c)$, $r_x$, $r_x^p$, and $r_x^c$.
>
> Figure 4 outlines our data-driven approach for sampling contexts and estimating LLM’s gender bias, contrasting it with traditional model-dependent or human-annotated methods. Words with yellow and blue backgrounds represent keywords filtered from the corpus, belonging to categories $X$ and $Y$, respectively. Sentences containing both categories are recorded. The blue arrows indicate the use of established NLP toolkits to parse these sentences, removing all extraneous content to extract essential sentence structures. The "C's Dist." column on the far right logs the frequency of these sentence structures, which are then used for subsequent risk calculations.
>
> **Measurement Value Scaling**
>
> Yes, $R$ and $R^P$ are of similar magnitude, while $R^v$ is much smaller. To shorten the width of Table 1, we presented $R^v$ in the current way, which seems to mislead readers. Thus, we agree with the reviewer and will present $R^v$ in a more direct way in the revision.
>
> **Results of Atypical Professions**
>
> We appreciate the reviewer's insightful suggestion and present the assessment of exceptional professions such as nurse and stylist, where the societal stereotype is pre-female. The attached PDF contains our assessment of gender discrimination for exceptional professions like nurse and stylist. Our assessment reveals two significant characteristics of the examined LLMs' gender bias in these professions: first, the LLMs exhibit a pro-male bias, contrary to societal stereotypes; second, the volatility risk associated with these professions is significantly higher than the model's average. These two characteristics are consistent across nearly all professions with a societal stereotype of being pre-female.
>
> These findings suggest that the LLMs' biases are the result of a complex mechanism. During the model training process, two types of information in the training data simultaneously influence the LLMs. One type is the pro-male societal stereotype related to the general notion of employment, and the other is the gender stereotype specific to each profession. Together, these two types of information determine the direction and volatility of the LLMs' gender bias regarding a profession. When a profession has a pre-female societal stereotype, these opposing gender-bias directions lead to prediction inconsistency and higher volatility risk. The direction of the LLMs' gender bias is contingent upon which type of information predominates during the training process. Our assessment indicates that the examined LLMs' training is predominantly influenced by the pro-male societal stereotype concerning employment.
>
> **Dataset Clarification**
>
> We employ the Toxic Tweets Dataset [Iyer, 2021], the Offensive Language Dataset [Davidson et al., 2017], and the Toxicity Dataset [Surge-ai, 2021] as our dataset for toxicity fine-tuning, which will be detailed in the main text. Due to space constraints, the dataset information for Tables 1 and 2 is in Appendix D, with plans to include excerpts in future versions. Additionally, references to the dataset for the results in section 4.1, now in Appendix D, will be explicitly mentioned.
>
> **X-Axis in Figure 9**
>
> The X-axis in Figure 9 shows the average hourly income for occupations in dollars. We will add the necessary labels in the revision.
>
> **Section 4.5 Significance**
>
> Section 4.5 explains and compares the fundamental nature of prejudice and volatility risks from a statistical theory perspective, providing valuable insights for future theoretical and technical studies on LLM alignment. Our analysis reveals that the prejudice risk for nearly all examined LLMs follows a normal distribution, while the volatility risk approximates a power law, resulting in a long-tail distribution.
>
> Theoretically, the findings in Section 4.5 suggest that while volatility risk may adhere to the neural network's scaling law, prejudice risk does not. Technically, these findings indicate that LLM alignment research should focus on developing hybrid training algorithms capable of simultaneously managing prejudice errors, which yield a normal distribution, and volatility errors, which result in a long-tail distribution. Additionally, we emphasize the importance of addressing prejudice and volatility risks separately in policy practice due to their distinct statistical characteristics. For instance, the long tail of volatility risk highlights the need for policymakers to focus on extreme events when deploying LLMs in real-world applications.
>
> Therefore, we propose to retain Section 4.5 and will enhance the writing to further clarify these points. However, if the reviewer still believes that Section 4.5 should be moved to the Appendix, we will consider your further recommendations.

---

> ### Author Rebuttal · Authors · 2024-08-17
>
> **Limitation Section**
>
> Due to space constraints, the limitation section is now located in Appendix F, focusing mainly on the exclusion of proprietary LLMs due to their undisclosed model prediction distribution details. We attach experiment results using multiple context template mining sources, including web pages, books, conversations, and textbooks, in the PDF. These results uniformly demonstrate the consistency of model discrimination risk rankings, affirming the robustness of the PVF measurement across various LLM application contexts. We will discuss data quality and representativeness in the revision.
>
> **Comparison with Existing Metrics**
>
> We provide a succinct summary of related work due to space limits. The study by [Gallegos, Isabel O., et al]. organizes current bias assessment metrics into three categories: embedding-based, probability-based, and generated text-based metrics. However, these methods primarily focus on whether the bias exhibited by models is systematic, overlooking the inconsistencies induced by contextual changes. We will extend discussion on related studies, and cite the survey by [Gallegos, Isabel O., et al].

---

### Official Review · Reviewer_aF2Z · 2024-07-25
**Review for Submission1404**

**Rating:** 6
**Confidence:** 3

**Review:**

See the below comments on strength, improvement and limitation of the study.

**Strengths:**

1-The framework is well defined.
2-The empirical evaluation of the framework on different open source models are well established.

**Additional Feedback:**

N/A

**Clarity:**

The paper is well written, however, the mathematical composition section of the paper could be simplified to improve accessibility.

**Correctness:**

It is constructed in a sound way to me. The framework is constructed well. The evaluation methods and experiment design are appropriate and performed well.

**Documentation:**

There is sufficient documentation. Reproducibility is assured.

**Ethics:**

Ethical concerns have been discussed. It should be expanded to include all the categories: (1) Research involving human subjects, (2) Data privacy, copyright, and consent, (3) Data quality and representativeness, (4) Safety and security, (5) Discrimination, bias, and fairness, (6) Deception and harassment, (7) Environmental Impact, (8) Human rights (including surveillance)

**Limitations:**

Authors have added a small section of limitations in the supplementary materials section. They only identify one limitation of the study and suggestion of mitigation. I think there are other limitations that the authors should have put into account and build mitigation strategies. such limitations are the following:
1- Focusing primarily on gender and racial biases, potentially overlooking other significant forms of discrimination
2- Generalization of the framework and capturing the nuances o how biases change across different contexts as well as different languages.
3- Using only Wikipedia dataset as a primary data source

**Opportunities For Improvement:**

1- Diversifying of the data source
2- Although the authors discussed education year and salary, Including a wider range of biases beyond gender and race would demonstrate the versatility and comprehensiveness of the PVF.

**Relation To Prior Work:**

The prior work is discussed in a short section. i would suggest to expand the section to clearly articulate the differentiate the framework.

**Summary And Contributions:**

The paper introduces the Prejudice-Volatility Framework (PVF) for quantifying biases in LLMs The proposed framework differentiates and quantifies discrimination in LLMs considering both their persistent biases (prejudices) and their preference changes across different contexts (volatility). The PVF is empirically tested on 12 language models in different sizes, revealing significant gender and racial biases. The authors employ a data-mining approach to approximate the various applied contexts of LLMs and develop statistical metrics to assess the resulting contextualized societal discrimination risk. The main contributions of the paper are the PVF that precisely defines behavioural metrics for assessing LLMs, mathematical decomposition of discrimination into prejudice and volatility risks, automated mining process for context templates from data sources, and benchmarking of 12 open source LLMs.

---

> ### Author Rebuttal · Authors · 2024-08-16
>
> Thank you for your feedback. We are delighted to know that you recognize the robustness of our discrimination assessment framework PVF, which breaks down discrimination into its components of prejudice and volatility, and that you find our empirical findings to be convincing. We would like to highlight the versatility of the PVF and clarify that the experiments demonstrated in our paper primarily serve to illustrate its utility in a specific context. While the methodology is broadly applicable across numerous scenarios, an exhaustive investigation of all such applications exceeds this paper's scope. Furthermore, it's worth noting that many aspects you refer to as limitations are indeed acknowledged in Appendices C and E, where we outline them as directions for future research. Your remaining questions are answered in sequence as they come.
>
> **Diversify Context Mining Data Sources**
>
> Thank you very much for your suggestions. Indeed, our PVF can be applied to assess the LLM’s discrimination risk for any dataset that corresponds to various scenarios in the LLM will be applied. One of the primary intentions behind designing PVF was to ensure that bias measurements can effectively adapt to different textual contexts. This can be achieved by applying the data mining methods from Section 3.2 to data from different sources.
>
> Following your comment, we use additional natural language datasets, including web pages, books, conversations, and textbooks, as sources for mining context templates per your suggestion. The results of the gender discrimination assessments, segmented by dataset, are presented in the attached PDF. We also document the context templates extracted and upload them to the code repository for further review.
>
> in the ranking of each model's risk level for gender bias, despite variations in the magnitude of measured risk across datasets. Specifically, models identified as the most biased consistently exhibit the highest levels of bias in all our test scenarios, and similarly, the least biased models maintain their ranking. This consistency suggests that the biases inherent in LLMs are uniform across application contexts.
> Furthermore, the variation in the magnitude of discrimination values across datasets highlights the necessity of pre-selecting contexts akin to those to which LLMs are exposed. We will incorporate those insights that are potentially helpful to regulatory practice into revision.
>
> **A Wider Range of Biases Measured**
>
> We would like to clarify that PVF can handle the measurement of discrimination prejudice and volatility involving attributes with more than two categories (i.e., $|Y| > 2$). The term for computing measurement values with any attribute number in practice is given by Equation 8. We present the assessment outcomes for binary gender in Table 1 and for quinary race in Table 2 in Appendix C.
>
> Additionally, as per your suggestion, we include measurements of a broader range of social discrimination, such as religion and disability, in the attached PDF. We will include them in the final version.
>
> **How Biases Change across Languages**
>
> The PVF framework can be applied to other languages. To manage various languages, it is necessary to redefine the demographic variable X with the attribute Y and subsequently obtain new context templates, similar to the process for different data sources.
> In the accompanying PDF, we present the measurement results of our model in Chinese. We found that models trained with both Traditional and Simplified Chinese texts exhibit greater gender bias compared to those trained exclusively on Simplified Chinese texts. While Simplified Chinese was developed and applied in the mainland after China’s revolution, we suggest the difference is caused by the cultural innovation associated with China’s historical progress, which deserves future studies.
>
> **Related Work Section**
>
> Due to space constraints, we provide only a succinct summary of related work. In our revision, we plan to expand this section to include a more comprehensive discussion of relevant studies. Stereotypes involve incorrect associations between specific demographic groups and certain characteristics. In the revised "Related Work" section, we will further explore various forms of these incorrect associations, including set-based associations in feature representations, probabilistic associations in model predictions, and performance-based associations in downstream tasks. However, existing methods primarily focus on whether the bias exhibited by models is systematic, often neglecting the inconsistencies that can arise due to contextual changes. Additionally, our approach can be applied to risk assessment across different textual contexts, offering a new perspective for interpretable alignment. We will incorporate these perspectives in our revision to further enrich the "Related Work" section.

---

### Official Review · Reviewer_yMyq · 2024-07-29
**This is an interesting paper that disentangle systemic and contextual biases in LLMs.**

**Rating:** 7
**Confidence:** 4
**Correctness:** Seems good overall.
**Clarity:** The paper is overall well written.

**Review:**

The novelty of the paper is in looking at two aspects of LLM's discrimination - prejudice (for systemic biases) and volatility (to measure bias inconsistencies based on context) and defining metrics for these. The evaluation also brings several interesting findings to light - such as
 - as models get larger, prejudice gets larger while inconsistencies decrease..
- RLHF reduces prejudice risk but increases volatility risk indicating that alignment does not entirely solve the problem.
- many models have pro-male biases indicating that societal biases are incorporated in the models.

Cons:

- The techniques only focus on binary gender, and it it not clear how this can be scaled to other biases.
- One thing that we can see is that where prejudice is lower, volatility is higher. It would be good to see what techniques could be used to balance this and have more meaningful models.
- How would the technique work when multiple biases are involved - such as gender + race ?
- More diverse datasets could be added to the analysis.

**Strengths:**

Please refer to the above answers

**Additional Feedback:**

NA

**Documentation:**

yes

**Limitations:**

- The work only looks at gender - other types of biases are unexplored and the scalability of technique and the efforts for dataset creation is not clear.
- In the real world, multiple biases are exhibited simultaneously - how do we handle this is not clear.

**Opportunities For Improvement:**

Mentioned in the cons category of the review.

**Relation To Prior Work:**

yes. The method brings in different aspects of evaluating bias/discrimination in LLMs (prejudice and volatility), novel metrics to identify these.

**Summary And Contributions:**

The paper presents a framework that they call the "Prejudice" and "Volatility" framework to evaluate biases or discrimination in LLMs. The paper tries to disentangle systemic and contextual biases in the models. Prejudice tries to identify the intrinsic biases in LLMs, while volatility brings out the variations or inconsistencies in LLM outputs in terms of biases, etc based on variation in contexts.

- The paper introduces metrics to measure and score the prejudice and volatility aspects of the LLM's biases and evaluates 12 different LLMs for their intrinsic and contextual biases.
- They evaluate across various aspects , such as size of the models, RLHF aligned vs not, etc..
- The framework also helps evaluate aspects such as salary vs gender - and understand the relation between LLM's discrimination risk and socio-economic factors.

The work is interesting and brings out important findings and points to further look into.

---

> ### Author Rebuttal · Authors · 2024-08-16
>
> We appreciate your insightful feedback and suggestions. We are glad to read that you find our social discrimination assessment framework PVF, which mathematically disentangles the models’ inherent learned biases (prejudice), and context-dependent biases (volatility), to be novel and bring out crucial findings. We arrange our responses in the order your questions are raised, and hope they clarify your concerns.
>
> **Broader Range of Biases Measured**
>
> We would like to clarify that the PVF framework is capable of measuring any form of discrimination that involves a finite number of attributes. Equation 8 theoretically explains how to manage different types of social discrimination. Consequently, PVF can be applied to assess both prejudice and volatility for discrimination issues involving more than two categories (i.e., |Y| > 2). For example, the results concerning LLMs' machinery race discrimination, which involve more than two types, are included in Table 2 of Appendix C.
>
> In response to your comment, we also examined a broader range of social discrimination, including those based on religion and disability. The results are summarized in Table 1 of the attached PDF. We will emphasize the generalizability of PVF across different types of social discrimination in our discussion of Equation 8 and will include the results of these additional types of discrimination in the results section.
>
> **Balancing Prejudice and Volatility**
>
> Thank you for your insightful question, which has prompted us to more comprehensively and deeply explain the contributions of our PVF approach to the future study of LLM alignment. Beyond offering an assessment method, the PVF also contributes to the following discussions:
>
> 1. Our discussion of the PVF illustrates that every LLM possesses a maximal degree of alignment achievable within its design constraints. Moreover, we introduce a method to calibrate a lower bound for this maximal alignment. From a statistical standpoint, each LLM can be viewed as a statistical estimator, where prejudice risk corresponds to bias, and volatility risk corresponds to variance. According to the Cramér-Rao bound, while bias can theoretically be eliminated, variance cannot. Consequently, the social discrimination risk inherent in LLMs cannot be entirely avoided, establishing the existence of a maximal achievable alignment. The extent to which an LLM’s discrimination declines when prejudice risk is minimized offers a lower bound for this maximal alignment.
> ﻿
> 2. Our PVF framework also suggests a novel perspective for developing alignment strategies for LLMs. Specifically, the PVF framework provides a tool for identifying the trade-off frontier between prejudice and volatility risks. The measures $R^P$ and $R^V$ can be utilized to craft reward or penalty functions within this context. By leveraging PVF-based alignment strategies, we can explore methods to optimally reduce the overall discrimination risk when deploying LLMs.
>
> Furthermore, PVF-based alignment method development can incorporate additional theories and methods from statistics. For example, the theory of minimum variance unbiased estimators (MVUEs) could be applied to the design of LLM alignment algorithms. In the revised version, we will emphasize the above contributions of our PVF approach, which we believe can inspire a range of future alignment studies from this new perspective.
>
> **Extension to Multiple Biases**
>
> As with the response to Question 1, our research can be applied to the evaluation of discrimination where $Y$ is discrete.
>
> When assessing biases that intertwine multiple attributes, such as "gender+race." This extension necessitates a precise definition of the evidence variable $X$, the attribute variable $Y$, and the selection of an appropriate context variable $C$ that meaningfully links $X$ and $Y$. For instance, to evaluate the mood states across various gender and racial groups, $X$ might encompass categories like "Black woman" and "Asian man," while $Y$ could represent mood states such as "happy" or "sad." In such scenarios, we can employ context templates outlined in our provided template/template\_mining\_n2a\_10000.csv file, for instance, "The [Y] [X] played a role." and "The [Y] [X] grew in popularity." Here, "[X]" is replaced with items from $X$, and we calculate the conditional probabilities for each mood category in $Y$, enabling the measurement of pertinent social discrimination. Overall, our PVF is versatile to quantify prejudice and volatility risk in LLMs, provided that $X$ and $Y$ are precisely defined, and $C$ is appropriately crafted to suit.
>
>
> **Diverse Context Mining Sources**
>
> We use additional natural language datasets, including web pages, books, conversations, and textbooks, as sources for mining context templates per your suggestion. The results of the gender discrimination assessments, segmented by dataset, are presented in Table 2 of the attached PDF. We also document the context templates extracted and upload them to the code repository for further review.
>
> Our findings indicate a notable consistency in the ranking of each model's risk level for gender bias, despite variations in the magnitude of measured risk across datasets. Specifically, models identified as the most biased consistently exhibit the highest levels of bias in all our test scenarios, and similarly, the least biased models maintain their ranking. This consistency suggests that the biases inherent in LLMs are uniform across application contexts.
>
> Furthermore, the variation in the magnitude of $R$ across datasets underscores that discrimination risk is scenario-dependent. When applying the PVF, it is crucial to select data that reflect the specific context of use. For instance, Llama 2 demonstrates a lower risk in scenarios involving textbook-related content compared to more general types of books. We plan to incorporate these insights into our revision, as they provide valuable guidance for regulatory practices.

---

### Decision · Program_Chairs · 2024-09-26

**Decision:**

Accept (Poster)

**Comment:**

This paper presents an interesting framework for "Prejudice" and "Volatility" via the evaluation of biases and discrimination in LLMs,aiming to disentangle both systemic and contextual biases in the models. They evaluate 12 different LLMs for their intrinsic and contextual biases with the framework and set the stage for further evaluation building upon this work.